# Effects of Loss and Gain Incentives on Adherence in Pediatric Weight Management: Preliminary Studies and Economic Evaluation of a Theoretical Trial

**DOI:** 10.3390/ijerph20010584

**Published:** 2022-12-29

**Authors:** Robert Siegel, Meghan E. McGrady, Linda Dynan, Roohi Kharofa, Kristin Stackpole, Paula Casson, Francesca Siegel, Nadine A. Kasparian

**Affiliations:** 1The Heart Institute, Cincinnati Children’s Hospital, Cincinnati, OH 45229, USA; 2Department of Pediatrics, College of Medine, University of Cincinnati, Cincinnati, OH 45267, USA; 3Division of Behavioral Medicine & Clinical Psychology, Cincinnati Children’s Hospital, Cincinnati, OH 45229, USA; 4James M. Anderson Center for Health Systems Excellence, Cincinnati Children’s Hospital, Cincinnati, OH 45229, USA; 5Department of Economics, Northern Kentucky University, Highland Heights, KY 41099, USA; 6The University of Cincinnati Medical Center, Cincinnati, OH 45219, USA

**Keywords:** childhood obesity, loss incentive, gain incentive, cost-effectiveness

## Abstract

Pediatric weight management is often hampered by poor engagement and adherence. Incentives based on loss have been shown to be more effective than gain-based incentives in improving outcomes among children with health conditions other than obesity. In preparation for a clinical trial comparing loss-framed to gain-framed incentives, a survey of youth and caregiver attitudes on weight management incentives, reasons for program attendance, and an economic evaluation of a theoretical trial were conducted. Ninety of 835 (11%) surveys were completed by caregiver and child. The economic evaluation showed that loss-framed incentives had a preferable incremental cost-effectiveness ratio (a lower value is considered preferable) than gain-based incentives. Most youth and caregivers felt a gain incentive would be superior, agreed that the full incentive should go to the youth (vs. the caregiver), and identified “improving health” as a top reason for pursuing weight management.

## 1. Introduction

Childhood obesity continues to be a major health issue, with its prevalence increasing at an alarming rate during the COVID-19 pandemic [1,2,3]. Pediatric weight management programs (PWMP) have been successful at improving the weight status of youths with obesity, but adherence and poor retention are common challenges [4,5]. Incentives have been helpful in improving adherence in both adults with health conditions and risky health behavior, such as smoking cessation, diabetes management, and obesity [6,7,8,9,10]. Additionally, incentives framed as a loss where individuals are given an incentive that loses value if a person does not adhere to a task are generally more effective than incentives framed as a gain where the incentive increases [11,12,13]. Gained-frame incentives in pediatric age groups have improved outcomes in the control of Type 1 Diabetes, improved food selection, and increased physical activity [14,15,16,17,18]. While loss appears to be a more powerful motivator in adolescents than gain, loss incentives have not been tested against gain-framed incentives in pediatric weight management [19].

Given the limited information on loss versus gain incentives, our group began planning a randomized controlled trial. With the preliminary studies described in this manuscript, we sought information on youth and caregiver attitudes toward incentives, reasons for weight management program attendance, and estimates of the cost-effectiveness of gain and loss-framed incentives.

## 2. Materials and Methods

This preliminary project was developed to gather data to assist with the design of a randomized controlled trial (RCT). It included two phases: (1) a survey to determine caregiver and youth attitudes about incentives and weight management, and (2) a cost-effectiveness analysis based on a theoretical study design using the survey results.

### 2.1. The Survey

An anonymous 24-item, investigator-created, caregiver-report survey and companion 15-item patient survey (Appendix A and Appendix B) were emailed to families of youth aged 12–18 years attending the Center for Better Health and Nutrition/Healthworks! PWMP between 1 April 2020 and 3 March 2021. The program has about 3000 patient visits per year. The survey was developed by a pediatrician with the American Board of Obesity Medicine Certification and two pediatric psychologists with expertise in adherence. Additionally, the survey was reviewed by members of the Childhood Obesity Multi-Program Analysis and Study System, a national pediatric practice-based network of about 25 pediatric weight management programs [20,21,22,23]. The questions were developed to help design how the incentives would be structured for a clinical trial comparing loss and gain-framed incentives as well as gaining more insight as to why families attend the program. Three email invitations to participate were sent in June 2021. The caregiver-report survey inquired about patient demographics (age, sex, race, ethnicity, healthcare insurance type, and caretaker-reported height and weight). Both caregiver- and child-report surveys included items assessing reasons for participating in lifestyle/weight management, attitudes regarding activities that should be rewarded, incentive amounts, how incentives would be portioned between caregiver and child, and whether how incentives were framed would increase PWMP adherence. The online survey was made available in English and Spanish via the secure REDCap platform. The study was approved by the Cincinnati Children’s Hospital Internal Review Board.

### 2.2. Theoretical Economic Evaluation

A decision analytic model was developed to compare the 6-month outcomes associated with three possible interventions: a control group (CG), a gain-framed incentive group (GG), and a loss-framed incentive group (LG). In the theoretical trial, all participants will receive a Fitbit tracker and internet-compatible electronic scale. Those in the control group will receive USD 10 per month regardless of participation. Youths in the GG and LG will receive a reward each month of a dollar amount equal to their adherence index. Those in the GG will have a virtual account that starts out at USD 0 each month and increases up to USD 100, depending on adherence. Those in the LG will have a virtual account that starts out at USD 100 and may decrease to as low as USD 0, depending on their adherence. The incentive scheme is summarized in Table 1.

The hypothetical cohort included 60 youth (aged 12 to 18 years) with obesity. Following randomization, youth were classified as “attending a 3-month visit” or “not attending a 3-month visit.” After 3 months, youth in both categories could be classified as “completed study” or “dropped out” (see Figure 1). The probability of transitioning to each of these states was estimated from previous literature based on adult data by Patel et al. in comparing loss versus gain incentives in improving physical activity in cardiac patients and from adherence data from pediatric weight management clinics without the use of incentives (Table 2) [24,25,26].

Health outcome. The health outcome of interest (denominator of the ICER) was adherence as measured via an adherence index developed to rate the level of participation of all those enrolled in the study with input from a dietitian, an exercise physiologist, physicians, and psychologists (Table 1).

Additional assumptions are listed in Table 2. Costs are summarized in Table 3, with costs of equipment obtained from the purchase invoice and estimates of medical, dietitian, and exercise physiologist obtained from Medicare reimbursement on Clearhealthcosts.com [27].

### 2.3. Statistical Analysis

For the survey, descriptive statistics and comparisons of caregiver and youth responses were carried out using IBM^®^ SPSS software (Armonk, NY, USA). For Likert scale responses, variables were dichotomized to 4, 5 vs. 1, 2, 3 and comparisons were made using chi-square analysis. For ranked order responses, differences were tested with Wilcoxon Rank Order analysis. Survey response data were dichotomized for questions with a Likert Scale response and the question on the reason for attending the program for ease of interpretation. A *P* value of less than 0.05 was considered significant. The effect size for differences between caretakers and youths was calculated as Eta-squared. A value of 0.01 to 0.059 was considered a small effect or association, 0.06 to 0.139 a medium effect, and greater than 0.14 a large effect. A sample size of 71 to 90 was determined using published criteria for a survey with a sampling error of 95% [28].

A decision tree for the study was created using TreeAge Pro 2022 software (Copyright 1988–2022 TreeAge Software, LLC, Williamstown, MA, USA) and the randomization process of the study protocol. The provider costs of visits 2, 4, and 5 were bundled together, as the probability of each of the visits was assumed to be equal for all branches in the analysis. A cost-effective analysis was estimated using the hypothesized values and one-way sensitivity analyses were performed varying the probability of adherence in the CG, LG, and GG by 10% higher and lower and for the maximum incentive with a range of USD 300 to USD 1200 for the entire study.

## 3. Results

### 3.1. Survey Results

In total, 90 of 835 (11%) surveys were returned. Most youths (55%) identified as Non-Hispanic White per caregiver report, with the remaining youth identifying as Black or African American (31%), Hispanic or Latino White (8%) and other (4%) per caregiver report. Sixty percent of families were insured through Medicaid. Youth’s mean age was 14.2 ± 1.8 years and mean percent of 95th percentile body mass index (BMI) was 128% ± 32%. Of the caregivers completing the survey, 88% were mothers, 3% were fathers, and 8% were other. Table 4 summarizes the results of the two top reasons listed by youths and their caregivers for attending the PWMP.

Most youth and their caregivers chose “improve health” as one of their top responses. Table 5 shows youth and caregiver responses to attitudes on potential incentives and framing approaches. Most youths and caregivers favored most or all the reward going to the youths and believed the incentive would motivate attending clinic, meeting exercise step goals, and food logging.

### 3.2. Economic Evaluation

Figure 1 shows the decision tree generated by TreeAge Software and Table 6 summarizes the costs, effectiveness, and cost-effectiveness of the three study strategies based on the assumptions made in Table 2. While the control group was the least expensive intervention, the loss group had a lower incremental cost-effectiveness ratio (ICER) than the gain group.

## 4. Discussion

With our study, we present preliminary data for developing a reward scheme for a study of loss- and gain-framed incentives for patients enrolled in a PWMP. Using assumptions based on adherence and loss and gain incentive studies of other conditions, we conducted a hypothetical cost-effectiveness analysis which suggests that loss incentives may lead to better adherence and result in lower costs per improvement in adherence.

According to the survey responses, youths preferred compensation rates of about USD 1 per day for meeting daily physical activity and food logging goals and USD 30 per month for clinic visit attendance. Both caregivers and youth preferred that all the rewards go to the child. These findings are in contrast to those reported by Wright et al. in which a survey of parents of children in PWMP suggested the reward should be split evenly between parent and child for PWMP [29]. Contrary to Prospect Theory, which suggests loss is a more powerful motivator than gain, most of the youth and parents in our study believed that gain incentives will be more effective than loss incentives [30]. Similar to our findings, parents in the survey reported by Wright favored a gain-framed incentive [29]. Consistent with our results, Goldsmith et al. in a series of lab experiments in undergraduates and adults, demonstrated that while loss incentives were more motivating than gain incentives, participants falsely predicted that gain incentives would be more motivating than loss incentives [31]. Goldsmith suggests that a biased belief in a positive correlation between enjoyment and task motivation is the cause of the participants’ incorrect prediction. Guided by our survey results, we designed the incentive to be up to USD 30 for a clinical visit, USD 1 for food log reporting, and USD 1 for meeting the step goal, with the entire incentive going to the youths participating in the economic analysis of the planned study. Additionally, of interest is that the majority of youths and caregivers in our study endorsed, “improve health” as one of the top two reasons for PWMP attendance. More youths than caregivers selected, “improve appearance”.

There are several limitations to our study. The survey response rate was only 11%, and results were derived from a single center, limiting generalizability and thus the results should be interpreted with caution. Additionally, weight, height, insurance status and race/ethnicity were self-report and we do not have comparison data for survey non-responders. A comparison with demographic data from a recent study from our center in 2020, suggests our sample had a larger percentage of white patients (55% versus 52%), and was somewhat higher socio-economic status based on a lower percentage of Medicaid insurance (60% versus 65%) [32]. Additionally, since the survey was anonymous, we do not have data reflecting program engagement such as the number of visits, length of time in the program, or show rate for clinic visits. It should be mentioned there is some discordance between caretaker results and their children. In constructing the incentive framework, we considered both youth and parent responses. With the economic evaluation of a simulated trial, there were multiple assumptions made that may not be the case in an actual clinical trial. Additionally, in the economic evaluation, the probabilities and visit costs were largely based on adult incentive studies and Medicare reimbursement data.

## 5. Conclusions

Youth with obesity participating in a PWMP and their caregivers report that incentives will motivate participation in all three major aspects of weight management. Contrary to behavioral economic theory, youth and their parents believe gain incentives are likely to be more effective than loss incentives. The results of our cost-effectiveness analysis suggest that loss-framed incentives may incur fewer costs per improvement in adherence. Additional research is needed to see if a loss or gain incentive is superior in improving adherence and to test the assumptions of our economic model. A randomized controlled trial comparing loss and gain-framed incentives to control is being conducted by our research group.

## Figures and Tables

**Figure 1 ijerph-20-00584-f001:**
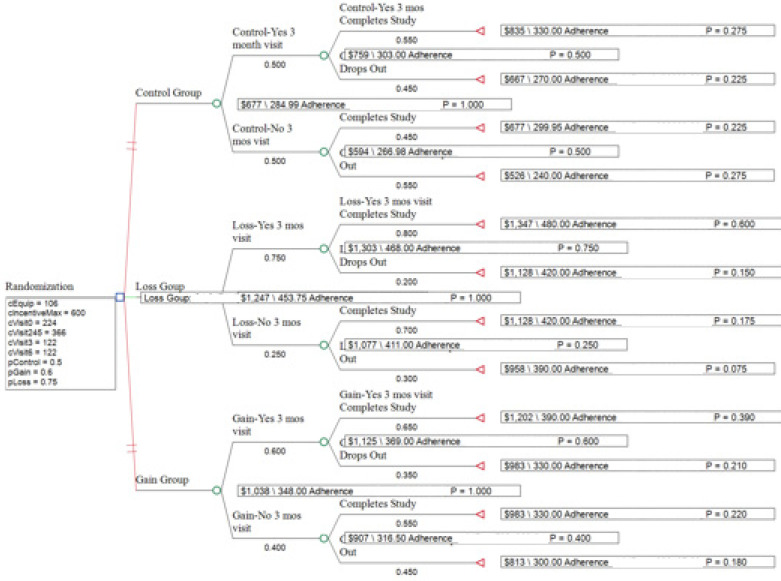
The TreeAge Diagram of the Study Strategies. Branches represent group randomization, the likelihood to keep the 3-month visit, and then completing study. Each branch has cost, adherence, and predicted probability.

**Table 1 ijerph-20-00584-t001:** Adherence Index/Incentive Scheme.

Type of Activity		When	Value	Adherence Index Points	Maximum Points per Month	Maximum Incentive per Month
Study Participation		Monthly	Yes or No	10 points	10 points	USD 10
Exercise Adherence	Daily steps by tracker	Daily	0 to 3000 steps	0 points per day	30 points	USD 30
			3001 to 5000 steps	0.5 points per day		
			5001 or more steps	1 point per day		
Dietary Adherence	Completing goal log answering questions: 1. Did you eat breakfast today? y/n (meal eaten before 10:30 am) 2. How many fruits or vegetables eaten today? 0 to >10 3. How many sugary drinks today? 0 to >10 4. Did you eat after 7 pm today? y/n	Daily	Yes or No	1 per day	30 points	USD 30
Clinical Adherence	In clinic (0,3,6 mos.) or Telehealth (1,2,4,5 mos.)	Monthly	Yes or No	30 points	30 points	USD 30
Maximum Total					100 points	USD 100

**Table 2 ijerph-20-00584-t002:** Definitions and assumptions made for hypothetical cost-effective analysis.

Parameter	Assumption/Definition
Cost	Cost = cost of equipment + cost of visits + cost of incentive. Visit costs are based on posted Medicare reimbursement for physician, dietitian, and exercise physiologist visits.
Adherence	The probability of task adherence is assumed to be uniform for all components (clinic visits, step goal, food reporting). The probability of adherence increases by 5% if the 3-month clinic visit is kept. It decreases by 5% if the visit is not kept. Estimates of adherence are based on baseline visit completion rates described in the literature and by our group’s previous rate of visit completion (50%) for the control group [25,26] Adherence estimates for GG and LG are based on changes report by Patel et al. in physical activity with gain (increase by 20%) and loss (increase by 50%) incentives [24]. The probability of keeping visits 2, 4, and 5 are uniform for each branch of the tree.
Drop out	“Drop out” is defined as not completing the 6-month clinic visit.
Sensitivity analysis range	Range is estimated as plus or minus 10% of adherence rate.
Utility	Utility is defined as the total adherence score totaled for all 6 study months.

**Table 3 ijerph-20-00584-t003:** Projected study costs.

Item	Cost	Comments
Equipment		
Fitbit	USD 66	
Scale	USD 40	
Provider Visits ***		
Initial Visit MD	USD 153	
Initial Visit Dietitian	USD 37	
Initial Visit Exercise Physiologist	USD 34	
Total cost of visit 1	USD 224	
Follow-up MD Visit	USD 51	
Follow-up Dietitian Visit	USD 37	
Follow-up Exercise Physiologist	USD 34	
Total Follow-up Visit	USD 122	
Cost of Visits 2, 4, 5	USD 366	
Maximum Incentive Study Incentive	USD 600	The actual incentive can vary dependent on the level of adherence as outlined in Table 1.
Study participation Incentive	USD 50 or	Dropout
(Control Group Only)	USD 60	Completes study

* From Medicare costs given on Clearhealthcosts.com.

**Table 4 ijerph-20-00584-t004:** Reasons for participation in PWMP (ranked either most or 2nd most important).

Reason	Youth Response	Caregiver Response	*Eta Squared ***	*p* Value *
Improve health	61.6%	97.6%	**0.0188**	**<0.0001**
Change weight	43.8%	39.0%	0.00640	0.791
Fitter or stronger	43.6%	50.6%	0.0011	0.537
Improve appearance	31.0%	7.3%	**0.0250**	**<0.0001**
Do not want to participate	9.8%	0.0%	**-**	**0.015**
Keep up with friends	8.4%	7.1%	0.00722	0.764
Meeting my goals	6.8%	Not an option	-	-

** By Chi-Square analysis, * *p* values significant at the 0.05 level indicated in bold typeface. Eta of medium or high effect, 0.06 or above is in bold.

**Table 5 ijerph-20-00584-t005:** Attitudes about incentives and overall health.

Question	Youth (Rating 4–5) % or Mean and SD	Caregiver (Rating 4–5) % or Mean and SD	*Eta-Squared*	*p* Value
Concerned about health	55.3%	84.5%	**0.043**	**0.03**
Reward will motivate clinic visit	71.2%	86.7%	**0.171**	**0.012**
Reward will motivate exercise goal	73.3%	78.9%	**0.0266**	**0.527**
Reward will motivate food log	71.1%	80.0%	**0.403**	**<0.0001** *
Amount for clinic incentive	USD 32.9 ± 17.9	USD 22 ± 16.5	**0.416**	**<0.001** *
Amount for food log	USD 0.75 ± 0.32	USD 0.80 ± 0.29	**0.348**	0.88 **
Amount for steps	USD 0.87 ± 0.27	USD 0.82 ± 0.32	0.00960	0.62 **
Ways to split (most or all to child)	66.7%	90.0%	0.121	**<0.001** *
Gain likely to motivate the most	53.0%	66.7%		
Loss likely to motivate most	6.7%	4.4%	**0.281**	**<0.001** *#
No preference Gain or Loss	23.1%	26.6%		

* Chi-Square, ** Wilcoxon Signed Rank. *p* values significant at the 0.05 level indicated in bold typeface, SD: standard deviation. Eta of medium or high effect, 0.06 or above is in bold. # Not dichotomized as there were three distinct categories.

**Table 6 ijerph-20-00584-t006:** Costs, Effectiveness, and Cost-Effectiveness of Loss, Gain, and Control Groups Relative to Control.

Strategy	Cost	Effectiveness (Adherence Points)	Cost/ Effectiveness	Incremental Cost	Incremental Effectiveness	ICER (IC/IE)
Control Group	USD 677	285	USD 2.38/point	-	-	-
Gain Group	USD 1038	348	USD 2.98/point	USD 361	63	5.73
Loss Group	USD 1247	454	USD 2.74/point	USD 570	169	3.38

## Data Availability

As per the policy of the Cincinnati Institutional Review Board, the survey data for this study is not publicly available.

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
