# Peer review of "Effects of Loss and Gain Incentives on Adherence in Pediatric Weight Management: Preliminary Studies and Economic Evaluation of a Theoretical Trial"

_ijerph, 2022, doi:10.3390/ijerph20010584_

Round 1

Reviewer 1 Report

The article has several weaknesses:

- To carry out the study, a survey is used. Was this survey agreed upon or reviewed by a group of experts?

- Although the authors mention as a limitation that only 11% of the surveys were completed by the caregiver and the children, considering a very small sample. We believe that this is not a sufficient sample to publish the article.

- In the statistical analysis, apart from the p-value, the effect sizes must be presented in tables and in the text with the interpretation of how significant is the relationship between the variables and/or the difference between groups.

- The discussion and conclusions are very brief. The repercussion or impact of the results in the clinical setting is not clear.

For all these reasons, we consider that the article should be rejected.

Author Response

Dear reviewer,

Thank you for the opportunity to revise and resubmit our manuscript, “Effects of Loss and Gain Incentives on Adherence in Pediatric Weight Management: Preliminary Studies and Economic Evaluation of a Theoretical Trial.”    We thank the reviewers for their insightful edits and questions and we believe that we have addressed all their concerns.   Our responses to the reviewers’ comments are below. 

Reviewer 1

Thank you for your critique and comments. 

The article has several weaknesses:

- To carry out the study, a survey is used. Was this survey agreed upon or reviewed by a group of experts?

Thank you for noting this was not clearly addressed.  This is now addressed in lines 57 to 63 with,

“The survey was developed by a pediatrician with the American Board of Obesity Medicine Certification and two pediatric psychologists with expertise in adherence to gain a greater understanding of youth and caretaker preference to help design how incentives will be designed for a study comparing Loss framed incentives to positive framed incentives.  Additionally, the survey was reviewed by members of Childhood Obesity Multi Program Analysis and Study System, a national pediatric practice-based network of about 25 pediatric weight management programs.[14-17]  .”

- Although the authors mention as a limitation that only 11% of the surveys were completed by the caregiver and the children, considering a very small sample. We believe that this is not a sufficient sample to publish the article.

Thank you for emphasizing this limitation.   This is now recognized more emphatically in lines 191 to 193 with,

“The survey response rate was only 11%, and results were derived from a single center, limiting generalizability and thus the results should be interpreted with caution.”

- In the statistical analysis, apart from the p-value, the effect sizes must be presented in tables and in the text with the interpretation of how significant is the relationship between the variables and/or the difference between groups.

Thank you for this suggestion.   The effect size is now shown with eta squared in Tables 4 and 5 and is described in the statistical analysis in lines 110 to 113, “The effect size for differences between caretakers and youths was calculated as Eta-squared.  A value of 0.01 to 0.059 was considered a small effect or association, 0.06 to 0.139 a medium effect, and greater than 0.14 a large effect.”

- The discussion and conclusions are very brief. The repercussion or impact of the results in the clinical setting is not clear.

For all these reasons, we consider that the article should be rejected.

Thank you for this suggestion.  The discussion section has now been expanded, lines 170-206,

“According to the survey responses, youths preferred compensation rates of about $1 per day for meeting daily physical activity and food logging goals, and $30 per month for clinic visit attendance. Both caregivers and youth preferred that all the reward go to the child. These findings are in contrast to those reported by Wright et al. (2019) in which a survey of parents of children in PWMP suggested the reward should be split evenly between parent and child for PWMP.[23] Contrary to Prospect Theory, which suggests loss is a more powerful motivator than gain, most of the youth and parents in our study believed that gain incentives will be more effective than loss incentives.[24] Similar to our findings, parents in the survey reported by Wright favored a gain-framed incentive.[23] Consistent with our results, Goldsmith et al. in a series of lab experiments in undergraduates and adults, demonstrated that while loss incentives were more motivating than gain incentives, participants falsely predicted that gain incentives would be more motivating than loss incentives.[25]  Goldsmith suggests that a biased belief in a positive correlation in enjoyment and task motivation is the a cause of the participants incorrect prediction.  Guided by our survey results, we designed the incentive to be up to $30 for a clinical visit, $1 for food log reporting, and $1 for meeting the step goal, with the entire incentive going to the youths participating in the economic analysis of the planned study. Also, of interest is that the majority youths and caregivers in our study endorsed, “improve health” as one of the top two reasons for PWMP attendance. More youths than caregivers selected, “improve appearance.”

There are several limitations to our study. The survey response rate was only 11%, and results were derived from a single center, limiting generalizability and thus the results should be interpreted with caution. Additionally, weight, height, insurance status and race/ethnicity were self-report and we do not have comparison data for survey non-responders.  A comparison with demographic data from a recent study from our center in 2020, suggests our sample had a larger percentage of white patients (55% versus 52%), and was somewhat higher socio-economic status based on a lower percentage of Medicaid insurance (60% versus 65%).[26]  Also, since the survey was anonymous, we do not have data reflecting program engagement such as the number of visits, length of time in the program, or show rate for clinic visits.    It should be mentioned there is some discordance between caretaker results and their children.  In constructing the incentive framework, we considered both youth and parent responses.    With the economic evaluation of a simulated trial, there were multiple assumptions made that may not be the case in an actual clinical trial.  Additionally, in the economic evaluation, the probabilities and visit costs were largely based on adult incentive studies and Medicare reimbursement data.  A clinical trial is now underway at our center that will test these assumptions.”

Please note there were value errors in Tables 4 and 5 that have been corrected.   Once again, many thanks to the editors and reviewers for their careful review and helpful suggestions.   Your efforts have led to a much better and more impactful manuscript.

Sincerely yours,

Robert Siegel, MD

Reviewer 2 Report

Overview and general recommendation:

I thank the authors for the opportunity to read and comment on their article. This manuscript is descriptive and presents preliminary survey results among parents and children regarding loss and gain weight loss incentives and theoretical cost-effectiveness analysis. The topic is of high importance and relevance. The methodology is interesting, presenting a theoretical nature. However, more statistical details are needed. My comments are detailed below.

2.1. Major comments:

Abstract

1.      Data on the representativeness of the sample and the survey's power should be added.

2.      Please clarify the phrase "preferable cost-effectiveness ratio."- ICER.

Introduction

Clear and suitable.

Methods

1.      A sample size calculation should be added.

2.      Please provide additional information on how the questionnaire was developed.

3.      Please clarify where were the BMI data obtained. Are they self-reported? Were they retrieved from the medical center?  

Results

1.      The authors describe low adherence to the survey (11%). Please supply information on the representability of the study population in major characteristics: age, sex, race, education, residency, etc.

2.     Figure 1 is graphically unclear; please improve it.

Discussion

1.      The discussion should mention the study's limitations.

2.      The fact that the study's results contradict what is known from experiments and knowledge of behavioral economics should be further discussed. The authors should provide theoretical explanations and try to find comparisons from the literature.

Language

The article is well written.

Author Response

Thank you for the opportunity to revise and resubmit our manuscript, “Effects of Loss and Gain Incentives on Adherence in Pediatric Weight Management: Preliminary Studies and Economic Evaluation of a Theoretical Trial.”    We thank the reviewers for their insightful edits and questions and we believe that we have addressed all their concerns.   Our responses to the reviewers’ comments are below. 

Reviewer 2

Overview and general recommendation:

I thank the authors for the opportunity to read and comment on their article. This manuscript is descriptive and presents preliminary survey results among parents and children regarding loss and gain weight loss incentives and theoretical cost-effectiveness analysis. The topic is of high importance and relevance. The methodology is interesting, presenting a theoretical nature. However, more statistical details are needed. My comments are detailed below.

2.1. Major comments:

Abstract

  1. Data on the representativeness of the sample and the survey's power should be added.

Thank you for this suggestion.   Given the length of abstract requirements, this is now included, but  in the methods for power and in the discussion for representativeness.

  1. Please clarify the phrase "preferable cost-effectiveness ratio."- ICER.

Thank you for this suggestion.  This is now made clearer in lines 20-22,

“The economic evaluation showed that loss-framed incentives had a preferable incremental cost-effectiveness ratio (a lower value is considered preferable) than gain-based incentives.”

Introduction

Clear and suitable.

Methods

  1. A sample size calculation should be added.

Thank you for pointing out this oversight.  A sample size estimate is now added on lines 113 to 114 with, “A sample size of 71 to 90 was determined using published criteria for a survey with a sampling error of 95%.

  1. Please provide additional information on how the questionnaire was developed.

Thank you for this suggestion.  This is now clarified in lines 57 to 63,

“The survey was developed by a pediatrician with the American Board of Obesity Medicine Certification and two pediatric psychologists with expertise in adherence to gain a greater understanding of youth and caretaker preference to help design how incentives will be designed for a study comparing Loss framed incentives to positive framed incentives.  Additionally, the survey was reviewed by members of Childhood Obesity Multi Program Analysis and Study System, a national pediatric practice-based network of about 25 pediatric weight management programs.” 

  1. Please clarify where were the BMI data obtained. Are they self-reported? Were they retrieved from the medical center?  

BMI was calculated from caretaker reported height and weight.  This is now clarified in lines 67 to 68,  “Three email invitations to participate were sent in June 2021. The caregiver-report survey inquired about patient demographics (age, sex, race, ethnicity, healthcare insurance type and caretaker-reported height and weight).”

Results

  1. The authors describe low adherence to the survey (11%). Please supply information on the representability of the study population in major characteristics: age, sex, race, education, residency, etc.

Thank you for this suggestion.  While unfortunately, we do not have the demographics from the cohort that received the invitation for this survey, we do have data from a similar time from our clinic.  This is now described in the discussion section in lines 195-198, “A comparison with demographic data from a recent study from our center in 2020, suggests our sample had a larger percentage of white patients (55% versus 52%), and was somewhat higher socio-economic status based on a lower percentage of Medicaid insurance (60% versus 65%).” 

  1. Figure 1 is graphically unclear; please improve it.

Figure 1 has been simplified by removing the variable box and there is now more explanation in the figure legend with, “Figure 1. The TreeAge Diagram of the Study Strategies.  Branches represent group randomization, the likelihood to keep a 3-month visit, and then completing the study.  Each branch has cost, adherence, and predicted probability.”

Discussion

  1. The discussion should mention the study's limitations.

The limitations paragraph of the Discussion has been greatly expanded, lines 191-205,

“There are several limitations to our study. The survey response rate was only 11%, and results were derived from a single center, limiting generalizability and thus the results should be interpreted with caution. Additionally, weight, height, insurance status and race/ethnicity were self-report and we do not have comparison data for survey non-responders.  A comparison with demographic data from a recent study from our center in 2020, suggests our sample had a larger percentage of white patients (55% versus 52%), and was somewhat higher socio-economic status based on a lower percentage of Medicaid insurance (60% versus 65%).[25]  Also, since the survey was anonymous, we do not have data reflecting program engagement such as the number of visits, length of time in the program, or show rate for clinic visits.    It should be mentioned there is some discordance between caretaker results and their children.  In constructing the incentive framework, we considered both youth and parent responses.    With the economic evaluation of a simulated trial, there were multiple assumptions made that may not be the case in an actual clinical trial.  Additionally, in the economic evaluation, the probabilities and visit costs were largely based on adult incentive studies and Medicare reimbursement data.”

  1. The fact that the study's results contradict what is known from experiments and knowledge of behavioral economics should be further discussed. The authors should provide theoretical explanations and try to find comparisons from the literature.

Thank you for this helpful suggestion.   Similarities and differences with existing literature are now included in the discussion section, lines 173-184,

“Both caregivers and youth preferred that all the reward go to the child. These findings are in contrast to those reported by Wright et al. in which a survey of parents of children in PWMP suggested the reward should be split evenly between parent and child for PWMP.[29] Contrary to Prospect Theory, which suggests loss is a more powerful motivator than gain, most of the youth and parents in our study believed that gain incentives will be more effective than loss incentives.[30] Similar to our findings, parents in the survey reported by Wright favored a gain-framed incentive.[29] Consistent with our results, Goldsmith et al. in a series of lab experiments in undergraduates and adults, demonstrated that while loss incentives were more motivating than gain incentives, participants falsely predicted that gain incentives would be more motivating than loss incentives.[31]  Goldsmith suggests that a biased belief in a positive correlation in enjoyment and task motivation is the a cause of the participants incorrect prediction.” 

Language

The article is well written.

Thank you.

Please note there were value errors in Tables 4 and 5 that have been corrected.   Once again, many thanks to the editors and reviewers for their careful review and helpful suggestions.   Your efforts have led to a much better and more impactful manuscript.

Sincerely yours,

Robert Siegel, MD

Reviewer 3 Report

This manuscript reports on findings from an interesting study that combines a survey of youth and caregivers regarding attitudes towards incentives for participation in a pediatric weight management program with an economic evaluation of the cost-effectiveness of employing gain and loss-framed incentives. The paper offers a novel perspective for considering the important challenge of limited engagement with, and attrition from, pediatric weight management programs. The manuscript is well-written and the work is clearly informed by relevant theoretical frameworks and empirical literature. Despite these clear strengths, there are some concerns with the manuscript as detailed in the following.

Introduction: This section is succinct and generally provides a good rationale for the study. It may be helpful to allocate a bit more space to “unpacking” the literature related to incentives, highlighting any work conducted with children and families. Doing so may help to provide justification for some of the assumptions made for the cost model and analysis.  

Methods: It would be helpful to provide additional rationale for the specific questions related to incentives that were included in the developed survey.

The authors provide information related to demographics of caregivers and youth who completed the survey. Given the scope of the questions asked, it would also be informative to include details related to their engagement with the pediatric weight management program. Specifically, it would be important to include any available information regarding their length of involvement with the program, attendance at scheduled visits, etc.

Table 2 includes a series of assumptions related to program adherence and cost. Some of these assumptions are based on studies with adults, which may not generalize to teens given the developmental differences and complexity of both teen and caregiver involvement. More generally, additional information is needed regarding the assumptions used to derive the hypothetical cost-effectiveness analysis provided in this table.

Results and Discussion: The rate of returned surveys is quite low, even for this type of survey research. While the authors acknowledge this as a limitation, this does not obviate the concern that the responses may not be representative of the patient population. This is particularly important in the current research, given the goal of addressing adherence and attrition in this population.

In analyzing data from the survey responses, the authors note that variables were dichotomized. It would be helpful to include a rationale for this decision.

A further consideration is the potential disconnect between caregiver and youth responses to a survey of opinions and actual behavior. This should be noted as a limitation.

It would be helpful to place the study findings in additional context - i.e. noting how the findings fit with existing literature. 

Author Response

Thank you for the opportunity to revise and resubmit our manuscript, “Effects of Loss and Gain Incentives on Adherence in Pediatric Weight Management: Preliminary Studies and Economic Evaluation of a Theoretical Trial.”    We thank the reviewers for their insightful edits and questions and we believe that we have addressed all their concerns.   Our responses to the reviewers’ comments are below. 

Review 3

This manuscript reports on findings from an interesting study that combines a survey of youth and caregivers regarding attitudes towards incentives for participation in a pediatric weight management program with an economic evaluation of the cost-effectiveness of employing gain and loss-framed incentives. The paper offers a novel perspective for considering the important challenge of limited engagement with, and attrition from, pediatric weight management programs. The manuscript is well-written and the work is clearly informed by relevant theoretical frameworks and empirical literature. Despite these clear strengths, there are some concerns with the manuscript as detailed in the following.

Introduction: This section is succinct and generally provides a good rationale for the study. It may be helpful to allocate a bit more space to “unpacking” the literature related to incentives, highlighting any work conducted with children and families. Doing so may help to provide justification for some of the assumptions made for the cost model and analysis.  

Thank you for this insightful critique.   We now make this clearer in the Introduction in lines 32 to 41, “Incentives have been helpful in improving adherence in both adults  with health conditions and risky health behavior, such as smoking cessation, diabetes management, and obesity.[6-10] Additionally, incentives framed as a loss where individuals are given an incentive that loses value if a person does not adhere to a task are generally more effective than incentives framed as a gain where the incentive increases. [11-13] Gained-frame incentives in pediatric age groups have improved outcomes in control of Type 1 Diabetes, improved food selection and increased physical activity.[14-18]   While loss appears to be a more powerful motivator in adolescents than gain, loss incentives have not been tested against gain-framed incentives in pediatric weight management.[19]

Methods: It would be helpful to provide additional rationale for the specific questions related to incentives that were included in the developed survey.

Thank you for this suggestion.  The rational for the survey questions is now clarified in the methods section in lines 63 to 65 with,  “The questions were developed to primarily help design how the incentives would be constructed for the clinical trial comparing loss and gain framed incentives as well as gaining more insight as to why families attend the program.”

The authors provide information related to demographics of caregivers and youth who completed the survey. Given the scope of the questions asked, it would also be informative to include details related to their engagement with the pediatric weight management program. Specifically, it would be important to include any available information regarding their length of involvement with the program, attendance at scheduled visits, etc.

These are excellent suggestions.  The demographics are described in the Results section lines 126 to 129.  Since the survey was anonymous, data on involvement and on non-responders is not available.   This is now mentioned in the limitations section as well as a comparison to known previously obtained demographic data which suggests a slightly smaller percentage of minorities in our survey with a somewhat higher rate of private insurance. This now appears in the limitations paragraph of the Discussion section, lines 195 to 198, “Additionally, weight, height, insurance status and race/ethnicity were self-report and we do not have comparison data for survey non-responders.  A comparison with demographic data from a recent study from our center in 2020, suggests our sample had a larger percentage of white patients (55% versus 52%), and was somewhat higher socio-economic status based on a lower percentage of Medicaid insurance (60% versus 65%).[25]  Also, since the survey was anonymous, we do not have data reflecting program engagement such as the number of visits, length of time in the program, or show rate for clinic visits.”

Table 2 includes a series of assumptions related to program adherence and cost. Some of these assumptions are based on studies with adults, which may not generalize to teens given the developmental differences and complexity of both teen and caregiver involvement. More generally, additional information is needed regarding the assumptions used to derive the hypothetical cost-effectiveness analysis provided in this table.

This is an excellent point about the data coming from adult studies and the novelty of the study of loss and gain incentives in the pediatric age group.   This along with the cost assumptions is now addressed in the limitations section in lines 202 to 205 with, “With the economic evaluation of a simulated trial, there were multiple assumptions made that may not be the case in an actual clinical trial.  Additionally, in the economic evaluation, the probabilities and visit costs were largely based on adult incentive studies and Medicare reimbursement data.” 

Results and Discussion: The rate of returned surveys is quite low, even for this type of survey research. While the authors acknowledge this as a limitation, this does not obviate the concern that the responses may not be representative of the patient population. This is particularly important in the current research, given the goal of addressing adherence and attrition in this population.

Thank you for this very validated criticism.  The limitations of the study based on response rate is further emphasized in lines 191 to 193, The survey response rate was only 11%, and results were derived from a single center, limiting generalizability and thus the results should be interpreted with caution.”

In analyzing data from the survey responses, the authors note that variables were dichotomized. It would be helpful to include a rationale for this decision.

Thank you for pointing out this oversight.  This is now addressed in the Methods section, Statistical Analysis, lines  108 to 110  with, “Survey response data was dichotomized for questions with a Likert Scale response and the question on the reason for attending the program for ease of interpretation.”

A further consideration is the potential disconnect between caregiver and youth responses to a survey of opinions and actual behavior. This should be noted as a limitation.

Thank you for this suggestion.  This is now recognized in the limitations paragraph of the Discussion section in lines 200 to 202 with, “It should be mentioned there is some discordance between caretaker results and their children.  In constructing the incentive framework, we considered both youth and parent responses.”

It would be helpful to place the study findings in additional context - i.e. noting how the findings fit with existing literature. 

Thank you for this excellent suggestion.   This is now addressed in the Discussion section lines 173-184, “Both caregivers and youth preferred that all the reward go to the child. These findings are in contrast to those reported by Wright et al. in which a survey of parents of children in PWMP suggested the reward should be split evenly between parent and child for PWMP.[23] Contrary to Prospect Theory, which suggests loss is a more powerful motivator than gain, most of the youth and parents in our study believed that gain incentives will be more effective than loss incentives.[24] Similar to our findings, parents in the survey reported by Wright favored a gain-framed incentive.[23] Consistent with our results, Goldsmith et al. in a series of lab experiments in undergraduates and adults, demonstrated that while loss incentives were more motivating than gain incentives, participants falsely predicted that gain incentives would be more motivating than loss incentives.[25]  Goldsmith suggests that a biased belief in a positive correlation in enjoyment and task motivation is the a cause of the participants incorrect prediction.”

Please note there were value errors in Tables 4 and 5 that have been corrected.   Once again, many thanks to the editors and reviewers for their careful review and helpful suggestions.   Your efforts have led to a much better and more impactful manuscript.

Sincerely yours,

Robert Siegel, MD

Round 2

Reviewer 3 Report

The authors have done an excellent job of responding to concerns raised in the initial review. Notably, they have provided additional background related to use of incentives in the Introduction and more context for understanding the findings in the Discussion section.

1. Additional information has been included related to development of the survey administered to caregivers and youth. This information is helpful but the added text should be reviewed for a lengthy sentence and missing words (lines 57-65).

2. In the second paragraph of the section titled Theoretical Economic Evaluation, the authors note that probabilities of study completion versus dropout are provided in Table 2. It may help the reader to understand this section better if a bit more information regarding how adherence estimates were generated was provided in the text and attribution to pediatric versus adults studies was clearly stated.  

Author Response

Thank you once again for your careful and insightful review of our manuscript and the opportunity to resubmit it.

The reviewer comments and our responses are below:

The authors have done an excellent job of responding to concerns raised in the initial review. Notably, they have provided additional background related to use of incentives in the Introduction and more context for understanding the findings in the Discussion section.

  1. Additional information has been included related to development of the survey administered to caregivers and youth. This information is helpful but the added text should be reviewed for a lengthy sentence and missing words (lines 57-65).

Thank you for this suggestion. This section has been edited and the redundancy removed. 

“The survey was developed by a pediatrician with the American Board of Obesity Medicine Certification and two pediatric psychologists with expertise in adherence.    Additionally, the survey was reviewed by members of Childhood Obesity Multi Program Analysis and Study System, a national pediatric practice-based network of about 25 pediatric weight management programs.[20-23]   The questions were developed to help design how the incentives would be structured for a clinical trial comparing loss and gain-framed incentives as well as gaining more insight as to why families attend the program. Three email invitations to participate were sent in June 2021. The caregiver-report survey inquired about patient demographics (age, sex, race, ethnicity, healthcare insurance type, and caretaker-reported height and weight).”

  1. In the second paragraph of the section titled Theoretical Economic Evaluation, the authors note that probabilities of study completion versus dropout are provided in Table 2. It may help the reader to understand this section better if a bit more information regarding how adherence estimates were generated was provided in the text and attribution to pediatric versus adults studies was clearly stated.  

Thank you for pointing out the lack of clarity here.   This is now addressed with, “The probability of transitioning to each of these states was estimated from previous literature based on adult data by Patel et al. in comparing loss versus gain incentives in improving physical activity in cardiac patients and from adherence data from pediatric weight management clinics without the use of incentives (Table 2).[24-26]

Thank you once again for all your help with this manuscript.

Sincerely,

Robert Siegel, MD